# THE DEFICIT OF NEW INFORMATION IN DIFFUSION MODELS: A FOCUS ON DIVERSE SAMPLES

## ABSTRACT

Diffusion models are renowned for their state-of-the-art performance in generating high-quality images. Identifying samples with new information beyond the training data is essential for data augmentation, especially for enhancing model performance in diverse and unforeseen real-world scenarios. However, the investigation of new information in the generated samples has not been well explored. Our investigation through the lens of information theory reveals that diffusion models do not produce new information beyond what exists in the training data. Next, we introduce the concept of diverse samples (DS) to prove that generated images could contain information not present in the training data for diffusion models. Furthermore, we propose a method for identifying diverse samples among generated images by extracting deep features and detecting images that fall outside the boundary of real images. We demonstrate that diverse samples exist in the generated data of diffusion models, attributed to the estimation of forward and backward processes, but it can only produce a limited number of diverse samples, underscoring a notable gap in their capabilities in generating diverse samples. In addition, our experiment on the Chest X-ray dataset demonstrates that the diverse samples are more useful in improving classification accuracy than vanilla-generated samples. The source code is available at `https://github.com/lypz12024/diffusion-diverse-samples`.

## 1 INTRODUCTION

Diffusion models (DMs) have recently gained significant attention for their capacity to generate high-quality samples. However, the potential of these models to generate samples containing more information than the original training data remains unexplored. If the generated samples merely replicate the information in the training data, their utility for augmenting datasets in downstream tasks could be limited. This limitation is particularly critical in the augmentation of limited data. Therefore, exploring whether generated samples can offer new and diverse information is crucial for advancing artificial intelligence (AI) capabilities in these domains. Efficient AI models require vast and varied datasets for optimal performance, and relying solely on limited training data can hinder their development. By generating high-quality images with rich new and useful information, researchers can create more robust and effective models. This research aims to explore the potential of diffusion models to produce such diverse samples. Our key contributions in this paper are threefold,

**Mathematical Analysis.** We rigorously analyze diffusion models using information theory, demonstrating that these models do not introduce new information beyond what training data contain. By examining entropy, mutual information, and Kullback-Leibler (KL) divergence, we show that the entropy of generated images closely aligns with that of the original data, indicating no additional information is created.

**Diverse Samples (DS) Metric.** We introduce diverse samples as a subset of generated images that contain more information and variability than the original training images. We propose a diverse sample metric to quantify the variability of synthetic images generated by diffusion models. Using deep features extracted from real and generated images, we identify diverse samples—those falling outside the boundary of real images. This metric offers a novel way of assessing the diversity of generated images.

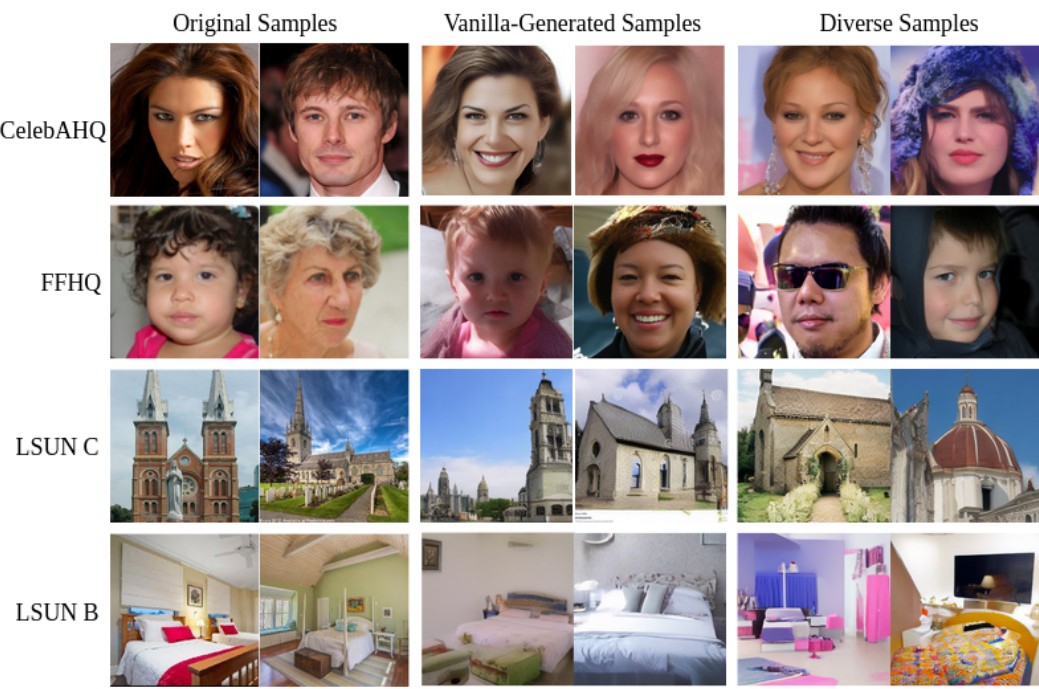

Figure 1: Comparison of original, vanilla generated, and diverse samples on different datasets. Vanilla-generated samples consist solely of generated images that do not include diverse samples. LSUN C denotes the LSUN Churches dataset Yu et al. (2015), and LSUN B refers to the LSUN Bedrooms dataset (Yu et al., 2015).

**Need for Better DS Generation Models.** We perform a comprehensive analysis of the performance of Latent Diffusion Models (LDMs) (Rombach et al., 2022) across four datasets with five solvers, uncovering significant limitations in their ability to generate diverse images. Our findings underscore the need for improved generative modeling techniques to enhance diversity, which is crucial for better data augmentation and more robust training data.

Fig. 1 illustrates examples of real images, vanilla-generated samples, and diverse samples, highlighting the distinct differences between vanilla-generated and diverse samples. The diverse samples exhibit more vibrant colors and greater feature variability. For instance, diverse samples from the CelebAHQ (Karras et al., 2017) and LSUN Bedrooms (Yu et al., 2015) datasets are notably more colorful than the vanilla-generated samples. In the case of the FFHQ dataset (Karras et al., 2019), diverse samples introduce additional features such as glasses and jackets over the hair, further emphasizing the enhanced variability present in these samples.

Our experiments reveal critical insights into the limitations and potential of diffusion models (DMs) for generating diverse samples. While DMs excel at replicating high-resolution images, their ability to produce genuinely diverse samples remains constrained. For instance, in the case of LSUN Churches dataset (Yu et al., 2015), certain solvers fail to generate any samples that deviate from the feature space of the original data as shown in Fig. 2 (a), highlighting a significant challenge in achieving true diversity. In contrast, Fig. 2 (b) reveals that some generated images contain new information, as they are positioned outside the distribution of the original real images. This suggests that the forward and backward diffusion process estimation allows for the generation of novel content not present in the training data. We also explored different performances of vanilla-generated samples and diverse samples in an imbalanced Chest X-ray dataset (Kermany et al., 2018). By incorporating diverse samples, we achieve substantial improvements in overall classification accuracy.

## 2 RELATED WORK

**Information Theory.** Shannon (1948) developed the foundational principles of information theory. Ali et al. (2022) first studied entropy in information theory from different perspectives and math-

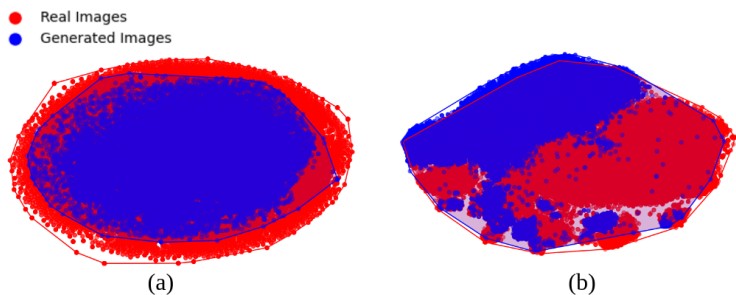

Figure 2: (a). t-SNE plot for deep features extracted by SqueezeNet (Iandola et al., 2016) model for real (red dots) and generated (blue dots) images on LSUN Churches dataset. The generated images are sampled with a DPM solver (Lu et al., 2022). This t-SNE plot reveals that the generated images are within the distribution of original real images, and no new information is generated from the diffusion model, as stated in our first contribution. (b). t-SNE plot showing deep features extracted by the EfficientNetB7 model on FFHQ dataset with DDIM generated images. This visualization demonstrates that some generated images contain new information positioned outside the distribution of the original real images, due to the forward and backward diffusion process estimation.

ematical models. Later, Ali et al. (2023) examined the AI notion of entropy and its applications in information theory. Tsalatsanis et al. (2021) proposed mutual information (MI), an information theory statistic, as a single measure to express diagnostic test performance. Uda (2020) provided an overview of the application of information theory to biological systems and discussed the associated bottlenecks. Zhang (2020) proposed a family of generalized mutual information whose members are indexed by a positive integer n, with the nth member being the mutual information of the nth order. Chen et al. (2021) proposed to integrate Shannon information theory with adversarial learning to accurately match visual and textual data in cross-modal retrieval.

**Diffusion Models and Solvers.** Diffusion models have gained considerable interest in image-generation tasks. Ho et al. (2020) pioneered diffusion probabilistic models, drawing from non-equilibrium thermodynamics to produce high-quality images. Nichol & Dhariwal (2021) made advancements to denoising diffusion probabilistic models (DDPMs) (Ho et al., 2020), introducing modifications that enhance sampling speed and log-likelihoods with minimal compromise on sample quality. Dhariwal & Nichol (2021) improved sample quality through classifier guidance, a method that leverages classifier gradients to efficiently balance diversity and quality. Song et al. (2020b) employed score-based generative modeling and numerical SDE (Stochastic Differential Equation) solvers for image generation. Additionally, Song et al. (2020a) developed denoising diffusion implicit models (DDIMs) to accelerate sampling while maintaining compatibility with the training procedures of DDPMs. Rombach et al. (2022) introduced latent diffusion models (LDMs), allowing diffusion models to be trained with limited computational resources while preserving quality and flexibility. Song et al. (2023) presented consistency models (CMs) that map noise to data in a single step, enabling multistep sampling that balances computational resources with sample quality. Lu et al. (2022) introduced the DPM-Solver, which generates high-quality samples with only 10 to 20 function evaluations. Zhao et al. (2024) proposed UniPC, a unified predictor-corrector framework for rapid DPM sampling, achieving synthesis in fewer than 10 inference steps. Zheng et al. (2024) developed the DPM-V3 solver, which samples images efficiently in 5 to 10 inference steps by incorporating several coefficients computed on the pre-trained model. Despite the progress made in diffusion models and solvers that improve image quality and reduce inference steps, a significant limitation persists: these models primarily focus on reproducing the training data rather than creating images that extend beyond it. They are designed to generate samples that closely resemble the training data, which often leads to limited diversity in outputs. Therefore, there is a need to enhance image quality and significantly boost the diversity of generated samples, enabling the creation of a wider array of images that go beyond the original dataset.

**Diffusion Models for Data Augmentation.** Diffusion models play a crucial role in data augmentation by enabling the generation of diverse images that can modify high-level semantic attributes, thereby overcoming the limitations of traditional augmentation techniques in enhancing data diversity. Trabucco et al. (2023), and Chen et al. (2024) used diffusion models for data augmentation to en-

hance the performance on real-world weed recognition tasks. Yao et al. (2023) proposed conditional diffusion model-based data augmentation for Alzheimer's prediction. Mueller (2024) proposed an attention-enhanced conditioning-guided diffusion-based approach for synthesizing additional training data to enhance machine fault diagnosis. Zhang et al. (2023) investigated single-image denoising diffusion model (SinDDM), and FewDDM, an extended version of SinDDM for medical image data augmentation with lung ultrasound images. Yu et al. (2023) proposed a diffusion-based augmentation method for nuclei segmentation in histopathology images. Zhong et al. (2024) proposed Meddiffusion for boosting health risk prediction via diffusion-based data augmentation. Fang et al. (2024) proposed data augmentation for object detection via controllable diffusion models and CLIP scores. Islam et al. (2024) proposed DIFFUSEMIX, a data augmentation technique that leverages a diffusion model to reshape training images, supervised by their bespoke conditional prompts. These authors utilized diffusion models for data augmentation to enhance performance in tasks such as image classification, object detection, and image segmentation. Although they demonstrated improved results compared to traditional augmentation techniques, the authors did not investigate the diversity of the generated images relative to the original training data and how the diverse sample can affect their results.

## 3 METHODS

### 3.1 INFORMATION THEORY

Information theory, founded by (Shannon, 1948), is a mathematical framework for quantifying information transmission, processing, and storage. Key concepts include: (1) **Entropy (H)**: Measures the uncertainty or randomness in a random variable or probability distribution; (2) **Mutual Information (I)**: Quantifies the amount of information obtained about one random variable through another. and (3) **Kullback-Leibler (KL) Divergence**: Measures the difference between two probability distributions.

### 3.2 DIFFUSION MODELS DO NOT GENERATE IMAGES WITH NEW INFORMATION

We use the concepts of entropy, mutual information, and KL divergence to show that images generated by diffusion models contain no new information compared to the training data.

**Entropy.** The entropy $H(X)$ of the training data $X$ ($x$ is one training data) is

$$H(X) = -\sum_x P_X(x) \log P_X(x). \tag{1}$$

The entropy $H(Y)$ of the generated data $Y$ ($y$ is one generated data)is

$$H(Y) = -\sum_y P_Y(y) \log P_Y(y). \tag{2}$$

**Mutual information.** The mutual information $I(X;Y)$ is defined as:

$$I(X;Y) = H(X) - H(X|Y), \tag{3}$$

where $H(X|Y)$ is the conditional entropy of $X$ given $Y$.

**Kullback-Leibler divergence.** The KL divergence $D_{KL}(P_X||P_Y)$ measures the differences between the training data distribution $P_X$ and the generated data distribution $P_Y$, which is defined as

$$D_{KL}(P_X||P_Y) = \sum_x P_X(x) \log \frac{P_X(x)}{P_Y(x)}. \tag{4}$$

If $P_X$ and $P_Y$ are identical, $D_{KL}(P_X||P_Y) = 0$.

**Information preservation in diffusion models.** Diffusion models aim to approximate $P_X$ with $P_Y$. For an ideal diffusion model $P_Y = P_X$. In this case, the KL divergence $D_{KL}(P_X||P_Y)$ is minimized by $D_{KL}(P_X||P_Y) = 0$. This implies that $P_X$ and $P_Y$ are identical. Since $P_Y = P_X$, the entropy of the generated images $H(Y)$ is approximately equal to the entropy of the training data $H(X)$:

$$H(Y) = H(X). \tag{5}$$

The mutual information $I(X;Y)$ is in an ideal case, when $P_X = P_Y$, that indicates that observing $Y$ provides almost all the information about $X$. We have

$$I(X;Y) = H(X). \tag{6}$$

Since $H(X|Y) = H(X) - I(X;Y)$ and $I(X;Y) = H(X)$, we get

$$H(X|Y) = 0. \tag{7}$$

This implies that an ideal conditional entropy $H(X|Y)$ is zero, meaning that there is no uncertainty in $X$ given $Y$. Thus, the generated images $Y$ do not introduce new entropy beyond what is present in the training data $X$. Hence, for an ideal diffusion model where the generated image distribution $P_Y$ is equal to the training data distribution $P_X$, the entropy of the generated images $H(Y)$ is approximately equal to the entropy of the training data $H(X)$. Therefore, the generated images do not contain new information beyond the training data.

### 3.3 NO INFORMATION GAIN DURING FORWARD AND BACKWARD PROCESSES

To further demonstrate that no new information is generated in the forward and backward processes of diffusion models (Ho et al., 2020; Song et al., 2020a; Rombach et al., 2022), we define the mutual information $I(x_0; x_T)$ to quantify the shared information between the original image $x_0$ and the noising image $x_T$.

**Forward process.** The forward process is typically defined as a sequence of transformations that gradually add noise to the data, making it more random over time. Mathematically, the forward process can be represented as:

$$q(\mathbf{x}_t|\mathbf{x}_{t-1}) = \mathcal{N}(\mathbf{x}_t; \sqrt{1 - \beta_t}\mathbf{x}_{t-1}, \beta_t\mathbf{I}), \tag{8}$$

where $\beta_t$ is the noise variance, and $\mathcal{N}$ denotes the normal distribution. For the forward process (noise addition), we have:

$$I(x_0; x_t) \leq I(x_0; x_{t-1}), \tag{9}$$

when the noise is added, the mutual information decreases since $x_t$ is more random and has less information about $x_0$.

**Reverse (Backward, Denoising) process.** The backward process seeks to reverse the effects of the forward process by gradually removing the noise to recover the original data. The backward process is given by:

$$p_\theta(\mathbf{x}_{t-1}|\mathbf{x}_t) = \mathcal{N}(\mathbf{x}_{t-1}; \mu_\theta(\mathbf{x}_t, t), \sigma_\theta^2(t)\mathbf{I}), \tag{10}$$

where $\mu_\theta$ and $\sigma_\theta$ are learned parameters that define the mean and variance of the reverse process at each step.

During the reverse denoising process, we have

$$I(x_0; x_t) \geq I(x_0; x_{t+1}), \tag{11}$$

as the noise is removed, the mutual information increases because $x_{t-1}$ carries more information about $x_0$. In an ideal case, the reverse process perfectly inverts the forward process. We have the following equation during the forward and the reverse processes:

$$I(x_0; x_T) \leq I(x_0; x_{T-1}) \leq \cdots \leq I(x_0; x_1) \leq I(x_0; x_0). \tag{12}$$

The ideal mutual information at the start and end should be the same:

$$I(x_0; x_0) = I(x_0; x_T). \tag{13}$$

Therefore, an ideal reverse process aims to make the distribution of generated samples as close as possible to the original data distribution, with no deviation, which represents no new information generated during the overall diffusion process. However, due to the approximation errors, the backward process cannot fully recover the original images, which implies some samples may lie outside the original data, which is proved in the following section.

### 3.4 Existence of New Information: Diverse Samples (DS) in the generated data

**What are Diverse Samples?** We introduce Diverse Samples (DS) as the subset of generated images from diffusion models that contain distinct and significant variations from the original training data. These diverse samples represent new information not captured by the training data. This concept is crucial for evaluating the diversity and potential utility of synthetic data, particularly in scenarios where augmenting datasets with truly varied examples is necessary for improving model performance. To show that DS exists, we demonstrate that $P_{\text{model}}$ cannot perfectly match $P_X$ in high-dimensional spaces, leading to some generated samples falling outside the distribution of the training data.

**KL divergence and perfect matching.** KL divergence $D_{KL}(P_X \| P_Y)$ quantifies the difference between the true data distribution $P_X$ and the model distribution $P_Y$:

$$D_{KL}(P_X \| P_Y) = \int P_X(x) \log \frac{P_X(x)}{P_Y(x)} \, dx. \tag{14}$$

If $P_X$ perfectly matches $P_Y$, then $D_{KL}(P_X \| P_Y) = 0$.

**High-Dimensional spaces and model limitations.** In high-dimensional spaces, it is challenging for generative models to perfectly capture the true data distribution. Diffusion models have finite capacity and cannot perfectly model complex, high-dimensional distributions. There are always some approximation errors in learning the true distribution. This implies:

$$\exists \, z \in Y \text{ such that } P_Y(z) \neq P_X(z). \tag{15}$$

Thus,

$$D_{KL}(P_X \| P_Y) > 0. \tag{16}$$

Because $P_Y$ cannot perfectly match $P_X$, there must exist regions in the data space where $P_Y$ assigns non-zero probability but $P_X$ assigns near-zero probability (or vice versa). These regions correspond to the diverse samples. Mathematically, this can be expressed as:

$$\exists \, z \in Y \text{ such that } P_X(z) \approx 0 \text{ and } P_Y(z) > 0. \tag{17}$$

Such samples $z$ are diverse because they belong to regions unlikely under the true data distribution but likely under the model distribution. Therefore, because of the high-dimensional probability distributions and the limitations of generative models, diffusion models cannot perfectly replicate the true data distribution. This imperfection inevitably leads to the generation of some DS.

### 3.5 Diverse Samples Calculation

We utilized original training images and their synthetic generated images from different solvers with pre-trained Latent Diffusion Models (LDMs) (Rombach et al., 2022). We extract deep features from the real and generated images using a pre-trained ImageNet model. Then, we visualize these features using t-SNE. By plotting the t-SNE representations, we define a boundary around the points corresponding to real images. Generated images whose t-SNE representations fall outside this boundary are identified as diverse samples. The process of calculating diverse samples follows the flowchart shown in Fig. 3 and is mathematically expressed as:

$$I_{\text{DS}} = \{I_{\text{gen},i} \mid \text{t-SNE}(f_d(I_{\text{gen},i})) \notin B_{\text{train}}, \forall i\}, \tag{18}$$

where $I_{\text{DS}}$ is the set of diverse samples, $I_{\text{gen}}$ is the set of generated images, $I_{\text{gen},i}$ is the $i$-th generated image, $f_d$ represents the deep features extracted from the images, $f_d(I_{\text{gen},i})$ represents the deep features extracted from the $i$-th generated image, $\text{t-SNE}(f_d(I_{\text{gen},i}))$ is the t-SNE representation of the deep features of the $i$-th generated image, and $B_{\text{train}}$ is the boundary of the t-SNE representations of the deep features of the training images.

The algorithm detailing our method for identifying diverse samples is provided in Appendix A. To create boundaries around the real images, we utilized the Quickhull algorithm (Barber et al., 1996) as outlined in Appendix A.

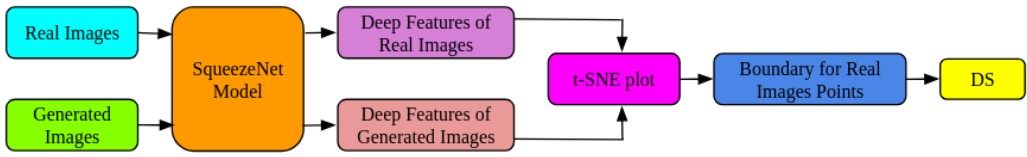

Figure 3: Flowchart for calculating Diverse Samples (DS).

# 4 RESULTS AND EXPERIMENTS

## 4.1 DATASETS

In our experiments, we utilize four well-known datasets and solvers to evaluate the diversity of images generated by Latent Diffusion Models (LDMs). The datasets include CelebAHQ (Karras et al., 2017), FFHQ (Karras et al., 2019), LSUN Churches (Yu et al., 2015), and LSUN Bedrooms (Yu et al., 2015), each offering a unique set of images for training and evaluation. CelebAHQ and FFHQ are comprised of high-quality human face images. The LSUN Churches and LSUN Bedrooms datasets, on the other hand, contain images of architectural interiors and exteriors, providing a different context for assessing the generative models. In addition, we utilize the Chest X-ray Images (Pneumonia) dataset (Kermany et al., 2018) to evaluate the impact of diverse samples on classification accuracy, imbalanced between the NORMAL and PNEUMONIA classes, to assess how augmenting the training data with diverse samples can influence the model's performance.

## 4.2 IMPLEMENTATION DETAILS

We generated 50,000 images with 8-step inference for each dataset with each solver using an A100 GPU. For generating these images, we utilized pre-trained weights of LDMs (Rombach et al., 2022) with a batch size of 100 and an eta ($\eta$) value set to 0. To generate the synthetic images, we employ various solvers, including DDIM (Song et al., 2020a), PLMS (Liu et al., 2022), DPM (Lu et al., 2022), UniPC (Zhao et al., 2024), and DPM-V3 (Zheng et al., 2024). To calculate diverse samples, we extracted deep features using pre-trained ImageNet models with a batch size of 500. These deep features are then used to plot t-SNE visualizations, which help us identify diverse samples by distinguishing those generated images that fall outside the feature space boundary defined by the real images. Additionally, for performing image classification on the Chest X-ray dataset, we employed a batch size of 128, the Adam optimizer, cross-entropy loss, and trained the ResNet50 (He et al., 2016) model for 20 epochs.

Table 1: The number of diverse samples in generated images on four datasets. We calculate these samples based on deep features extracted by the SqueezeNet model. We generate 50,000 samples for each of the datasets with five solvers.

| Dataset | Solver | # DS |
|---|---|---|
| CelebAHQ | DDIM (Song et al., 2020a) | 61 |
| | PLMS (Liu et al., 2022) | 67 |
| | DPM (Lu et al., 2022) | 54 |
| | UniPC (Zhao et al., 2024) | 42 |
| | DPM-V3 (Zheng et al., 2024) | 37 |
| FFHQ | DDIM (Song et al., 2020a) | 58 |
| | PLMS (Liu et al., 2022) | 53 |
| | DPM (Lu et al., 2022) | 78 |
| | UniPC (Zhao et al., 2024) | 39 |
| | DPM-V3 (Zheng et al., 2024) | 49 |
| LSUN Churches | DDIM (Song et al., 2020a) | 8 |
| | PLMS (Liu et al., 2022) | 8 |
| | DPM (Lu et al., 2022) | 0 |
| | UniPC (Zhao et al., 2024) | 8 |
| | DPM-V3 (Zheng et al., 2024) | 2 |
| LSUN Bedrooms | DDIM (Song et al., 2020a) | 3 |
| | PLMS (Liu et al., 2022) | 1 |
| | DPM (Lu et al., 2022) | 12 |
| | UniPC (Zhao et al., 2024) | 1 |
| | DPM-V3 (Zheng et al., 2024) | 0 |

## 4.3 RESULTS

We present our results in Table. 1, revealing interesting patterns in the samples generated across four different datasets and five solvers. A key observation is the complete absence of diverse samples in

Table 2: Chest X-ray dataset and classification accuracy comparison. OI Acc denotes classification accuracy with original images, VGS Acc denotes classification accuracy with vanilla-generated samples, and DS Acc denotes classification accuracy with diverse samples. The last column shows the change in accuracy with diverse samples over original images.

| Class | # Train | # Test | OI Acc | VGS Acc | DS Acc |
|---|---|---|---|---|---|
| NORMAL | 1341 | 234 | 0.4359 | 0.4957 | **0.5983** (↑ **16.24%**) |
| PNEUMONIA | 3875 | 390 | 0.9974 | 0.9974 | 0.9897 (↓ 0.77%) |
| OVERALL | 5216 | 624 | 0.7869 | 0.8093 | **0.8429** (↑ **5.60%**) |

the case of LSUN Churches with the DPM solver, where not a single generated image fell outside the boundary defined by the real images' t-SNE representations. This suggests that the DPM solver, when applied to the LSUN Churches dataset, generates images almost entirely confined within the feature space of the training data, indicating a lack of novelty or variation in the generated images. Similarly, there is a complete absence of diverse samples in the LSUN Bedrooms dataset with the DPM-V3 solver. Moreover, the overall number of diverse samples is relatively low for both LSUN Churches and LSUN Bedrooms across all solvers. For LSUN Churches, the number of diverse samples ranges from 2 to 8, and for LSUN Bedrooms, it ranges from 1 to 12. These low counts indicate that the generated images for these datasets predominantly mimic the real images, with less deviation from the established feature boundaries. In contrast, the CelebAHQ and FFHQ datasets exhibit more diversity in the generated samples. For CelebAHQ, the number of diverse samples identified ranges from 37 with the DPM-V3 solver to 67 with the PLMS solver. For the FFHQ dataset, the range varies from 39 to 78. These findings demonstrate that although diffusion models are proficient at replicating the training data, their capacity to generate genuinely novel images is limited. This highlights the challenges in achieving diversity when the generative process closely aligns with the original data distribution. The scarcity of diverse samples observed in our experiments points to a significant opportunity for improvement in generative modeling, particularly for data augmentation. To enhance the effectiveness of data augmentation, it is crucial to increase the diversity of generated samples. This may involve refining existing solvers or developing new techniques that promote greater variability in outputs, ultimately leading to more comprehensive and effective training data for downstream applications.

**Evaluation of diverse samples on image classification.** In this study, we assess the impact of diverse samples (DS) on image classification using the Chest X-ray Images (Pneumonia) dataset (Kermany et al., 2018), which is imbalanced between the NORMAL and PNEUMONIA classes. The dataset size and corresponding accuracy results are presented in Table. 2. Initially, we trained a ResNet50 (He et al., 2016) model on the original dataset and observed that the accuracy for the minority class, NORMAL, is relatively low at 43.59%, while the accuracy for the majority class, PNEUMONIA, is high at 99.74%. This results in an overall accuracy of 78.69%. To address the class imbalance, we augment the training set for the NORMAL class with vanilla-generated samples (VGS), matching the number of images in the PNEUMONIA class. This augmentation increases the accuracy for the NORMAL class by 5.98 percentage points, reaching 49.57%, and slightly improves the overall accuracy by 2.24 percentage points, raising it to 80.93%.

We then replace the vanilla-generated samples with diverse samples to introduce more variability in the training data. This approach significantly enhances the accuracy for the NORMAL class by 16.24%, raising it to 59.83%, and boosts the overall accuracy by 5.60%, increasing it to 84.29%. These results highlight the effectiveness of incorporating diverse samples in mitigating class imbalance, leading to improved classification performance and better model generalization, particularly for the underrepresented class. However, the accuracy for the PNEUMONIA class slightly decreases when using diverse samples, from 99.74% with the original images and vanilla-generated samples to 98.97% with diverse samples. This reduction in accuracy could be attributed to more variability and potentially noisier samples that, while beneficial for improving the classification of the minority class, may increase the complexity of the decision boundary for the majority class. This trade-off emphasizes the need for careful consideration when augmenting data with diverse samples, particularly in cases where class balance and overall accuracy are critical.

**How to generate diverse samples for this experiment?** For generating diverse samples, we employ a brute-force approach, where we generate a large number of samples and then filter out the diverse ones based on our method described in Fig. 3. This process is rigorous and requires us to continuously generate samples until we achieve the necessary quantity of diverse samples. The demanding nature

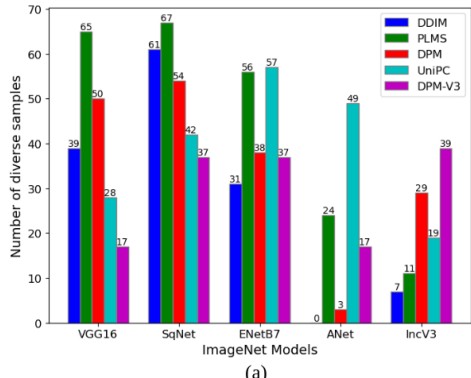 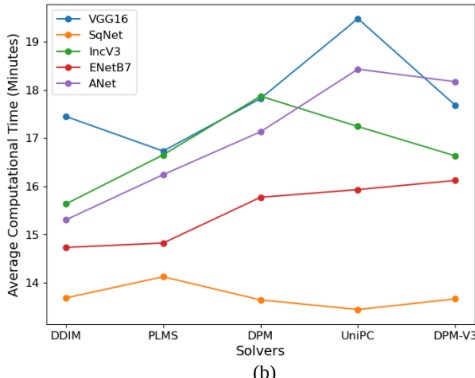

(a)                          (b)

Figure 4: (a). The number of diverse samples calculated from the 50,000 generated images of the CelebAHQ dataset. These samples are determined based on deep features extracted using various ImageNet models. SqNet refers to the SqueezeNet model, ENetB7 to the EfficientNetB7 model, ANet to the AlexNet model, and IncV3 to the InceptionV3 model. (b). Comparison of average computational time taken by different ImageNet models for calculating diverse samples across 50,000 images generated by several solvers.

of this method highlights the need for developing improved generative modeling techniques that can more efficiently and effectively produce diverse samples.

**Ablation study** In this section, we explore the impact of different ImageNet models on the number of diverse samples using the CelebAHQ dataset. As shown in Fig. 4 (a), the number of diverse samples identified by five pre-trained ImageNet models VGG16 (Simonyan & Zisserman, 2014), SqueezeNet (Iandola et al., 2016), EfficientNetB7 (Tan & Le, 2019) AlexNet (Krizhevsky et al., 2012), and InceptionV3 (Szegedy et al., 2016) demonstrates a consistent trend. While the number of diverse samples varies among five solvers, the differences across models are minimal, suggesting that the choice of pre-trained model does not substantially affect the number of diverse samples identified.

These results underscore that our method for calculating diverse samples is robust and unaffected by the choice of pre-trained model or solver. For instance, while SqueezeNet and EfficientNetB7 identify a higher number of diverse samples in several cases, this number is still very small compared to the 50,000 generated samples. This highlights the limited proportion of diverse samples within the overall generated dataset. The overall number of diverse samples remains relatively stable across different solvers, indicating that our approach effectively identifies diverse samples regardless of these variations. Additional results on other datasets are available in Appendix E, further validating the generalization and reliability of our method. We also calculate the time taken by different ImageNet models, as shown in Fig. 4 (b), to find an efficient model for calculating diverse samples. The SqueezeNet model shows consistently lower computation times across all solvers, ranging from approximately 13.44 to 14.12 minutes. On the other hand, VGG16 shows higher computation times, ranging from 16.73 to 21.48 minutes. The EfficientNetB7, InceptionV3, and AlexNet models exhibit higher computation times compared to SqueezeNet but lower times compared to VGG16. Therefore, we choose SqueezeNet as the most efficient model for calculating diverse samples.

## 5 CONCLUSION

In this paper, we find that diffusion models excel at generating high-resolution images, but they primarily replicate the information found in the training data, offering limited diversity. We prove that no new information is generated in an ideal diffusion model. By introducing the concept of diverse samples (DS) and using deep feature extraction with boundary-based analysis, we calculate the rare instances where generated images differ from the training data, highlighting the restricted diversity that these models currently achieve. Our experiments on the Chest X-ray dataset demonstrate the effectiveness of augmenting the data with diverse samples in improving classification accuracy. Our findings underscore the need for future research to improve diffusion models in generating more diverse samples that extend beyond the informational boundaries of the training data.

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

## A    ALGORITHMS

Alg. 1 shows the details of our method for identifying the diverse samples. We applied the Quickhull algorithm (Barber et al., 1996) to create boundaries for real images. Alg. 2 shows the details for generating the boundary for real images.

---

**Algorithm 1** Algorithm for Identifying Diverse Samples

---

**Input:** Training images $I_{\text{train}}$, Generated images $I_{\text{gen}}$, Deep features $f_d$, t-SNE parameters $p_{t-SNE}$
**Output:** Diverse samples $I_{\text{DS}}$
**Initialization:**
Extract deep features: $F_{\text{train}} \leftarrow f_d(I_{\text{train}})$
Extract deep features: $F_{\text{gen}} \leftarrow f_d(I_{\text{gen}})$
Apply t-SNE on training features: $T_{\text{train}} \leftarrow \text{t-SNE}(F_{\text{train}}, p_{t-SNE})$
Apply t-SNE on generated features: $T_{\text{gen}} \leftarrow \text{t-SNE}(F_{\text{gen}}, p_{t-SNE})$
Compute boundary of real images: $B_{\text{train}} \leftarrow \text{Boundary}(T_{\text{train}})$
**Identify Diverse Samples:**
**For each** $t_{\text{gen}} \in T_{\text{gen}}$ **do**
    **If** $t_{\text{gen}} \notin B_{\text{train}}$ **then**
        Add corresponding image to diverse samples: $I_{\text{DS}} \leftarrow I_{\text{DS}} \cup \{\text{corresponding image of } t_{\text{gen}}\}$
**End For**
Return $I_{\text{DS}}$

---

**Algorithm 2** Quickhull Algorithm for Constructing the Boundary of Real Images

---

**Input:** Set of points $P$ representing t-SNE features of real images
**Output:** Boundary $B$ for real images
**Initialization:**
Find the point with minimum x-coordinate: $A \leftarrow \min_{p \in P}(p_x)$
Find the point with maximum x-coordinate: $B \leftarrow \max_{p \in P}(p_x)$
Partition the set of points:
    $S_1 \leftarrow \{p \in P \mid p \text{ is to the left of line } AB\}$
    $S_2 \leftarrow \{p \in P \mid p \text{ is to the right of line } AB\}$
**Recursive Boundary Construction:**
**Function** FindBoundary($S, P, Q$):
    **If** $S$ is empty **then**
        **Return** []
    Find the point $C$ in $S$ that is furthest from line $PQ$
    Partition the set $S$ into two subsets:
        $S_1 \leftarrow \{p \in S \mid p \text{ is to the left of line } PC\}$
        $S_2 \leftarrow \{p \in S \mid p \text{ is to the left of line } CQ\}$
    **Return** FindBoundary($S_1, P, C$) + [$C$] + FindBoundary($S_2, C, Q$)
**End Function**
**Compute Boundary:**
$B \leftarrow [A]$ + FindBoundary($S_1, A, B$) + [$B$] + FindBoundary($S_2, B, A$)
Return $B$

---

## B    MORE DIVERSE SAMPLES (DS)

Fig. 5 presents more diverse samples, comparing them to the original and vanilla-generated samples from the CelebAHQ (Karras et al., 2017), FFHQ (Karras et al., 2019), LSUN Churches (Yu et al., 2015), and LSUN Bedrooms (Yu et al., 2015) datasets. The diverse samples display a broader range

of colors and increased variability in features. For example, in the CelebAHQ and FFHQ datasets, the diverse samples include additional features like caps over the hair and glasses for the eyes. In the LSUN Churches dataset, the diverse samples exhibit features such as trees surrounding the churches, adding more depth to the scenes. Similarly, in the LSUN Bedrooms dataset, the diverse samples are notably more colorful compared to the vanilla-generated samples.

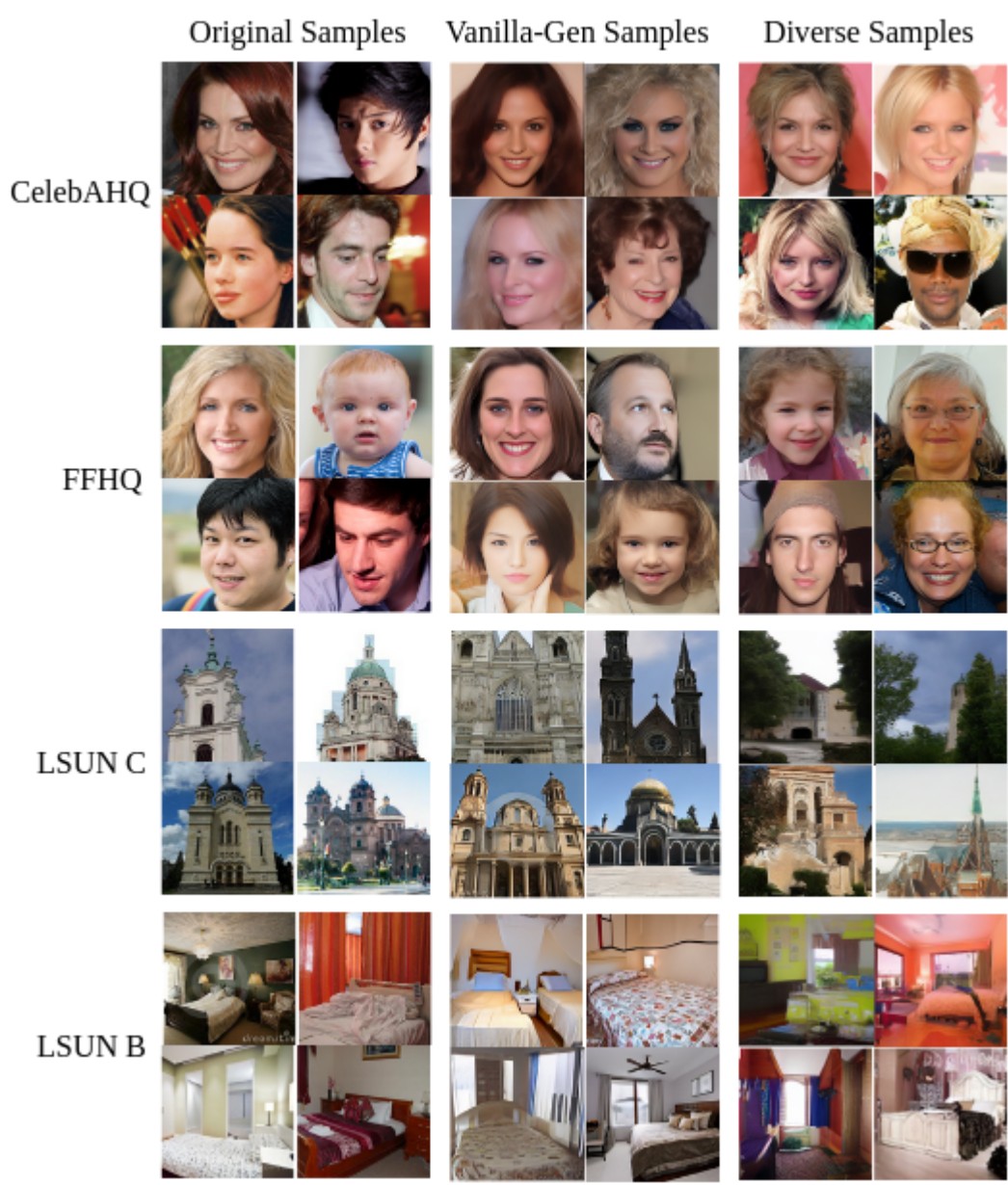

Figure 5: Comparison of original, vanilla generated, and diverse samples on different datasets. Vanilla-generated samples consist solely of generated images that do not include diverse samples. LSUN C denotes the LSUN Churches dataset (Yu et al., 2015), and LSUN B refers to the LSUN Bedrooms dataset (Yu et al., 2015).

## C  MORE T-SNE PLOTS

In Fig. 6, we present t-SNE visualizations of deep features extracted by four different models-SqueezeNet (SqNet) (Iandola et al., 2016), EfficientNetB7 (ENetB7) (Tan & Le, 2019), AlexNet

(ANet) (Krizhevsky et al., 2012), and InceptionV3 (IncV3) (Szegedy et al., 2016)-highlighting the relationship between real and generated images across different datasets and solvers. It demonstrates that the generated images remain within the distribution of the original real images, indicating that no new information is produced by the diffusion model.

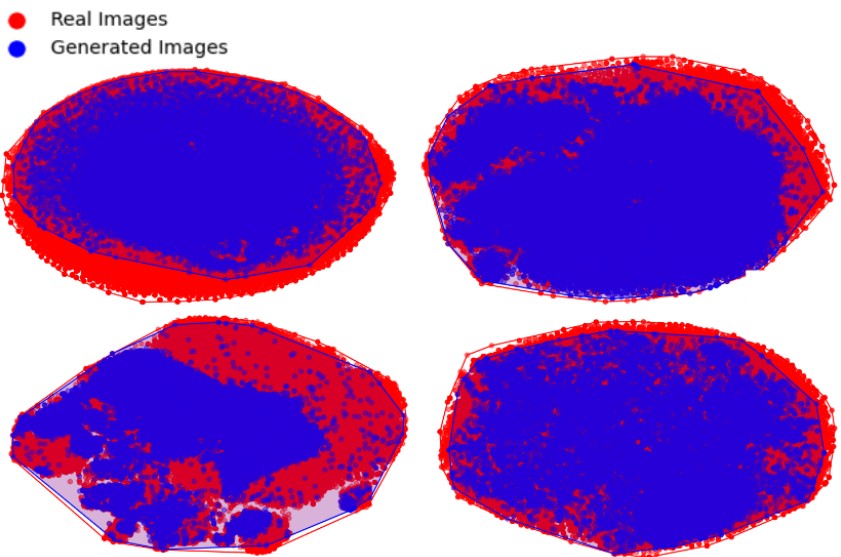

Figure 6: t-SNE plots for deep features extracted by the SqueezeNet, EfficientNetB7, AlexNet, and InceptionV3 models for real (red dots) and generated (blue dots) images on CelebAHQ, FFHQ, and LSUN Bedrooms datasets. The top left plot corresponds to the LSUN Bedrooms dataset with DDIM generated images, the top right corresponds to CelebAHQ with images sampled from the DDIM solver, the bottom left corresponds to FFHQ with UniPC sampled images, and the bottom right corresponds to LSUN Bedrooms with images generated from the DPM solver. These t-SNE plots reveal that the generated images are within the distribution of original real images, and no new information is generated from the diffusion model, as stated in our first contribution.

## D    COMPARISON OF THE NUMBER OF DIVERSE SAMPLES DETERMINED BY FIVE DIFFERENT IMAGENET MODELS

The results in Table 3 provide a comprehensive comparison of the number of diverse samples identified by five different ImageNet models—VGG16 (Simonyan & Zisserman, 2014), SqueezeNet (SqNet) (Iandola et al., 2016), EfficientNetB7 (ENetB7) (Tan & Le, 2019), AlexNet (ANet) (Krizhevsky et al., 2012), and InceptionV3 (IncV3) (Szegedy et al., 2016)—across four datasets: CelebAHQ (Karras et al., 2017), FFHQ (Karras et al., 2019), LSUN Churches (Yu et al., 2015), and LSUN Bedrooms (Yu et al., 2015). Each model's ability to detect diverse samples was evaluated using five different solvers, with a total of 50,000 generated samples per solver, employing an 8-step inference process. For the CelebAHQ dataset, SqueezeNet consistently identified the highest number of diverse samples across multiple solvers, with a peak of 67 samples using the PLMS solver. In contrast, AlexNet detected significantly fewer diverse samples, with no samples identified under the DDIM solver. Notably, the DPM solver also revealed a lower performance for all models, with AlexNet identifying only 3 diverse samples and VGG16 detecting 50 samples. EfficientNetB7 and InceptionV3 detected moderately fewer samples across all solvers, with EfficientNetB7 identifying 31 samples under DDIM and 57 under UniPC. In the FFHQ dataset, a stark difference in performance is observed, particularly with the EfficientNetB7 model, which detected 1,909 diverse samples using the DDIM solver—a figure far exceeding that of the other models. This higher detection rate in FFHQ can be attributed to the rich variability present in the FFHQ dataset, including a wide range of facial features, accessories, and lighting conditions. This inherent diversity in the dataset itself provides a broader spectrum of features for the models to detect and classify, leading to a higher number of identified diverse samples. In contrast, other models, such as VGG16 and SqueezeNet,

detected far fewer samples, with VGG16 identifying 198 diverse samples and SqueezeNet detecting 58 under DDIM. The DPM-V3 solver shows similar patterns, with EfficientNetB7 detecting only 1 sample, highlighting variability in model sensitivity. For LSUN Churches, the number of diverse samples is relatively low across all models and solvers. VGG16 identified 32 samples under PLMS, while SqueezeNet and EfficientNetB7 consistently detected fewer samples across all solvers, with SqueezeNet identifying only 8 samples under PLMS and DDIM. Interestingly, AlexNet performed comparatively better, identifying 26 samples under DDIM and 14 under DPM. These results highlight the challenges of generating diverse samples in this dataset, as the generated images seem more constrained within the training data's feature space. Similarly, in the LSUN Bedrooms dataset, solvers such as DDIM, UniPC, and DPM-V3 resulted in no or few diverse samples detected by SqueezeNet, EfficientNetB7, or InceptionV3. The DPM solver produced slightly better results, with EfficientNetB7 detecting 23 diverse samples. However, other solvers, such as PLMS and DDIM, showed limited diversity detection across models, with most models identifying fewer than 10 diverse samples. These findings underscore the challenge of generating truly novel or varied images in these datasets.

Table 3: The number of diverse samples in generated images on four datasets. We calculate these samples based on deep features extracted by the five ImageNet models. SqNet refers to the SqueezeNet model, ENetB7 to the EfficientNetB7 model, ANet to the AlexNet model, and IncV3 to the InceptionV3 model. We generate a total of 50,000 samples with each solver using an 8-step inference process.

| Dataset | Solver | # VGG16 | # SqNet | # ENetB7 | # ANet | # IncV3 |
|---|---|---|---|---|---|---|
| CelebAHQ | DDIM (Song et al., 2020a) | 39 | 61 | 31 | 0 | 17 |
| | PLMS (Liu et al., 2022) | 65 | 67 | 56 | 24 | 11 |
| | DPM (Lu et al., 2022) | 50 | 54 | 38 | 3 | 29 |
| | UniPC (Zhao et al., 2024) | 28 | 42 | 57 | 49 | 19 |
| | DPM-V3 (Zheng et al., 2024) | 17 | 37 | 37 | 17 | 39 |
| FFHQ | DDIM (Song et al., 2020a) | 198 | 58 | 1909 | 37 | 341 |
| | PLMS (Liu et al., 2022) | 212 | 53 | 4 | 51 | 71 |
| | DPM (Lu et al., 2022) | 172 | 78 | 417 | 53 | 105 |
| | UniPC (Zhao et al., 2024) | 178 | 39 | 0 | 18 | 85 |
| | DPM-V3 (Zheng et al., 2024) | 106 | 49 | 1 | 38 | 340 |
| LSUN Churches | DDIM (Song et al., 2020a) | 4 | 8 | 3 | 26 | 2 |
| | PLMS (Liu et al., 2022) | 32 | 8 | 6 | 7 | 6 |
| | DPM (Lu et al., 2022) | 0 | 0 | 1 | 14 | 1 |
| | UniPC (Zhao et al., 2024) | 31 | 8 | 10 | 6 | 2 |
| | DPM-V3 (Zheng et al., 2024) | 2 | 2 | 1 | 4 | 3 |
| LSUN Bedrooms | DDIM (Song et al., 2020a) | 0 | 3 | 6 | 1 | 0 |
| | PLMS (Liu et al., 2022) | 1 | 1 | 3 | 3 | 1 |
| | DPM (Lu et al., 2022) | 1 | 12 | 23 | 1 | 0 |
| | UniPC (Zhao et al., 2024) | 1 | 1 | 1 | 0 | 2 |
| | DPM-V3 (Zheng et al., 2024) | 1 | 0 | 1 | 6 | 0 |

# E    COMPARISON OF THE COMPUTATIONAL EFFICIENCY OF FIVE DIFFERENT IMAGENET MODELS

Since the differences between the number of diverse samples detected by different models are minimal compared to the total 50,000 generated images, we prioritize selecting the model based on computational efficiency. The computational time comparison across different models as shown in Table. 4, reveals that SqueezeNet consistently requires the least time to calculate diverse samples for all datasets, making it the most efficient model in this context. On the other hand, VGG16 takes the most time, with the computational time for EfficientNetB7, AlexNet, and InceptionV3 models falling between those of SqueezeNet and VGG16. When comparing the time taken across different datasets, CelebAHQ has the lowest computational time, while LSUN Bedrooms exhibits the highest. This variation in computational time across datasets can be attributed to the number of training images

Table 4: Comparison of average time taken (in minutes) to calculate diverse samples on four popular datasets based on deep features extracted by the five ImageNet models. We generate a total of 50,000 samples with each solver using an 8-step inference process (SqNet: SqueezeNet, EB7: EfficientNetB7).

| Dataset | Solver | VGG16 | SqNet | EB7 | AlexNet | InceptionV3 |
|---|---|---|---|---|---|---|
| CelebAHQ | DDIM (Song et al., 2020a) | 17.45 | 13.68 | 14.73 | 15.30 | 15.63 |
| | PLMS (Liu et al., 2022) | 16.73 | 14.12 | 14.82 | 16.24 | 16.65 |
| | DPM (Lu et al., 2022) | 17.82 | 13.64 | 15.77 | 17.13 | 17.87 |
| | UniPC (Zhao et al., 2024) | 21.48 | 13.44 | 15.93 | 18.43 | 17.24 |
| | DPM-V3 (Zheng et al., 2024) | 17.68 | 13.66 | 16.12 | 18.17 | 16.63 |
| FFHQ | DDIM (Song et al., 2020a) | 22.75 | 20.50 | 22.07 | 21.73 | 21.77 |
| | PLMS (Liu et al., 2022) | 22.00 | 19.80 | 21.50 | 20.85 | 21.54 |
| | DPM (Lu et al., 2022) | 23.01 | 20.23 | 21.42 | 22.05 | 21.40 |
| | UniPC (Zhao et al., 2024) | 21.42 | 21.56 | 22.58 | 22.60 | 21.87 |
| | DPM-V3 (Zheng et al., 2024) | 23.01 | 20.86 | 22.78 | 21.85 | 22.08 |
| LSUN Churches | DDIM (Song et al., 2020a) | 39.12 | 31.75 | 38.02 | 33.20 | 36.97 |
| | PLMS (Liu et al., 2022) | 36.72 | 31.03 | 34.27 | 35.03 | 35.67 |
| | DPM (Lu et al., 2022) | 37.23 | 32.33 | 37.01 | 32.71 | 36.87 |
| | UniPC (Zhao et al., 2024) | 38.42 | 35.07 | 38.50 | 38.16 | 37.32 |
| | DPM-V3 (Zheng et al., 2024) | 39.66 | 34.43 | 38.03 | 34.66 | 37.18 |
| LSUN Bedrooms | DDIM (Song et al., 2020a) | 97.05 | 81.15 | 93.64 | 86.98 | 96.12 |
| | PLMS (Liu et al., 2022) | 105.95 | 83.11 | 104.05 | 97.07 | 106.05 |
| | DPM (Lu et al., 2022) | 110.46 | 94.24 | 94.22 | 108.80 | 103.43 |
| | UniPC (Zhao et al., 2024) | 115.28 | 85.20 | 110.88 | 97.22 | 101.47 |
| | DPM-V3 (Zheng et al., 2024) | 107.63 | 77.90 | 102.35 | 105.83 | 105.40 |
| | Mean time | 46.54 | **37.88** | 43.93 | 43.20 | 44.46 |

Table 5: Core Notations

| | |
|---|---|
| $x; X$ | One training data; all training data |
| $y; Y$ | One generated data; all generated data |
| $P_X$ | Probability distribution of the training data |
| $P_Y$ | Probability distribution of the generated data |
| $H(X)$ | Entropy of the training data |
| $H(Y)$ | Entropy of the generated data |
| $I(X; Y)$ | Mutual information between the training data and the generated data |
| $D_{KL}(P_X \| P_Y)$ | KL divergence between the training data distribution and the generated data distribution |
| $X_0$ | Initial training data |
| $X_t$ | Training Data at time step $t$ in the forward process |
| $T$ | Final time step |
| $q(X_t | X_{t-1})$ | Forward process transition probability |
| $p(X_{t-1} | X_t)$ | Reverse process transition probability |

considered for calculating diverse samples: 30,000 for CelebAHQ, 60,000 for FFHQ, 119,916 for LSUN Churches, and 287,969 for LSUN Bedrooms.

## F    CORE NOTATIONS

The notations used throughout the Methods section are summarized in Table. 5.

