# OpenReview forum: "The Deficit of New Information in Diffusion Models: A Focus on Diverse Samples"
_ICLR.cc/2025/Conference — ICLR 2025 Conference Withdrawn Submission_

### Official Review · Reviewer_XFFr · 2024-10-25

**Soundness:** 2
**Presentation:** 3
**Contribution:** 2
**Rating:** 3
**Confidence:** 4

**Summary:**

This paper examines whether diffusion models generate new information beyond training data, finding that they typically replicate existing information. To address this, the authors introduce Diverse Samples (DS)—generated images that differ from the training data due to model approximations. By identifying these samples through feature analysis, the study shows that diverse samples enhance classification accuracy in underrepresented classes, such as in a Chest X-ray dataset, underscoring the value of improved diversity in diffusion models for effective data augmentation.

**Strengths:**

1. The question of whether and to what extent diffusion models can generate samples with new information is both interesting and important.
2. The experiments seem easy to reproduce, as the authors have provided the source code.

**Weaknesses:**

1. **The mathematical analysis is not very sound**: Equation (13) states that “the ideal mutual information at the start and end should be the same,” which doesn’t make sense to me. Since $x_T$ is pure Gaussian noise, it shouldn’t carry any information relevant to the image $x_0$.
2. **The arguments are not very convincing**: The authors argue that ideal diffusion models perfectly align with the training distribution and, therefore, cannot introduce new information, whereas imperfections allow the model to generate diverse samples that deviate from the training data manifold. However, the paper uses a pre-trained latent diffusion model, which could be trained on data that includes the four datasets used. The generation of diverse samples does not imply they contain “information not present in the training data.”
3. **Experimental results**: The results on the Chest X-ray dataset using diverse samples (DS) are not very compelling (e.g., the DS accuracy for the PNEUMONIA class is lower than that of the original images). To substantiate the claim that diverse samples selected by the proposed algorithm can enhance accuracy, additional experiments on different datasets are needed, rather than focusing on just one dataset.

**Questions:**

See the questions in the above.

---

### Official Review · Reviewer_8Jjp · 2024-10-29

**Soundness:** 2
**Presentation:** 2
**Contribution:** 1
**Rating:** 3
**Confidence:** 4

**Summary:**

This paper presents an information-theoretic analysis of diffusion models, demonstrating that they usually fail to generate truly novel information beyond their training data. In fact, theoretically perfect diffusion models are shown to produce no new information at all. To address this limitation, the authors introduce the concept of a *Diverse Sample* (DS). A DS is a sample generated by the diffusion model that contains information not present in the training data. By filtering generated samples to identify DSs, the authors propose a method to augment datasets and improve the performance of downstream tasks like classification.

**Strengths:**

1. This paper raises a critical concern about the effectiveness of generative model-based data augmentation for downstream tasks. It argues that generated samples may not contain information distinct from the original dataset, potentially limiting their ability to improve performance.
2. The proposed DS-based augmentation technique demonstrates significant performance improvements on imbalanced chest X-ray (CXR) classification tasks compared to traditional generative model-based approaches.
3. The paper's concepts and algorithms are clear and easy to follow.

**Weaknesses:**

1. The paper's first claim regarding the limitations of diffusion models in generating new information during training and inference is a well-established fact in the field of generative modeling. This is not a novel contribution. All generative models, by design, aim to learn and reproduce the underlying data distribution.
2. The proposed *Diverse Sample* (DS) method bears significant similarity to the technique employed in the *improved Precision-Recall* [1] method to identify out-of-distribution samples. While the DS method relies on a less robust approach, potentially making it more susceptible to noise and outliers compared to the KNN-based method in [1].
3. The assertion that DS samples contain novel, augmentation-worthy information is questionable. Given the nature of generative models, it is more likely that these samples represent low-probability, low-quality outputs that the model has incorrectly generated. The decreasing trend of DS samples with improved samplers, as evident in Table 1, further supports this argument.
4. The paper's analysis of generative model-based data augmentation diverges from the prevailing understanding. Data augmentation is typically employed when training data is a limited subset of the true data distribution. The effectiveness of this technique stems from the inherent generalization capabilities of machine learning algorithms [2]. Generative models are expected to approximate the true distribution, enabling the generation of synthetic data that can enhance model performance. The paper's perspective does not align with this widely accepted view.

[1] Kynkäänniemi, Tuomas, et al. "Improved precision and recall metric for assessing generative models." *NeurIPS* (2019).

[2] Chen, Yunhao, Zihui Yan, and Yunjie Zhu. "A comprehensive survey for generative data augmentation." *Neurocomputing* (2024): 128167.

**Questions:**

1. Is data augmentation using DS effective in balanced datasets? Is it also effective in general image classification tasks beyond chest X-ray analysis?
2. How does the performance of data augmentation using DS compare to methods employing diffusion models for minority sample generation, such as the approach in [3]?
3. Please provide a proof for equations (9) and (10). Does this property still hold true when using a scheduler, such as the one proposed in [3], that ensures zero signal-to-noise ratio (SNR) at the final timestep $T$?
4. The identified weakness of the proposed method is its susceptibility to incorrect sampling, which can result in the generation of erroneous data samples. Are there techniques to address this limitation?

[3] Um, Soobin, et al. "Don't Play Favorites: Minority Guidance for Diffusion Models." *ICLR* (2024).

[4] Lin, Shanchuan, et al. "Common diffusion noise schedules and sample steps are flawed." *WACV* (2024).

---

### Official Review · Reviewer_jXy5 · 2024-11-03

**Soundness:** 4
**Presentation:** 2
**Contribution:** 2
**Rating:** 3
**Confidence:** 5

**Summary:**

The paper explores diffusion models’ ability on generating out-of-distribution samples, which is an essential aspect for data augmentation. By calculating t-SNE on the extracted features from both real and generated images, it is stated that diffusion models rarely generate out-of-distribution data.

**Strengths:**

1. The paper is motivated by an essential question of whether a diffusion model can generate diverse samples, and therefore, can serve as a reliable data augmentation technique.

2. The paper reveals the limitation of diffusion models on generating out-of-distribution data through both information theory and experiments.

**Weaknesses:**

1. Evaluating the diffusion models’ ability on generating out-of-distribution data seems not to be an unexplored area, taking these works as examples [1][2]. Thus, there is concern about the paper’s novelty. It would be better if the author can include these previous work and discuss the difference.

2. The dimension of t-SNE seems not explicitly mentioned in the paper. As a dimensional reduction technique, I hardly agree only considering the top significant features can tell if the generated samples are out-of-distribution or not. Difference in less significant features can also lead to out-of-distribution data. I would suggest evaluating the correlation between the number of diverse samples and the number of dimensions used in t-SNE.

[1] Ramachandran, Sai Niranjan, et al. "Understanding the Generalization of Pretrained Diffusion Models on Out-of-Distribution Data." Proceedings of the AAAI Conference on Artificial Intelligence. Vol. 38. No. 13. 2024.

[2] Graham, Mark S., et al. "Denoising diffusion models for out-of-distribution detection." Proceedings of the IEEE/CVF Conference on Computer Vision and Pattern Recognition. 2023.

**Questions:**

My main concern is that if the diffusion model cannot generate out-of-distribution data, I am curious why the generated images from the diffusion model can lead to performance boost in classification tasks on Chest X-ray. I would suggest the examination on whether the performance boost comes from increased data volume or if there are subtle variations in the generated images that contribute to improved classification.

---

### Note · Authors · 2024-11-25

I have read and agree with the venue's withdrawal policy on behalf of myself and my co-authors.